# Plant Species Complementarity in Low-Fertility Degraded Soil

**DOI:** 10.3390/plants11101370

**Published:** 2022-05-21

**Authors:** Zhang Wei, Thomas Maxwell, Brett Robinson, Nicholas Dickinson

**Affiliations:** 1Faculty of Agriculture and Life Sciences, Lincoln University, Lincoln, Christchurch 7647, New Zealand; wei.zhang@lincolnuni.ac.nz (Z.W.); tom.maxwell@lincoln.ac.nz (T.M.); 2Department of Chemistry, University of Canterbury, Christchurch 8140, New Zealand; brett.robinson@canterbury.ac.nz

**Keywords:** soil nutrients, plant nutrition, co-existence, rhizosphere, phosphorus, manganese

## Abstract

The aim of this study was to investigate the compatibility of plants with contrasting root systems, in terms of procurement of limiting soil nutrients. Paired combinations of species of proteas and grasses were grown in a pot experiment using soil from a site with impoverished vegetation and degraded soil. The soil contained sufficient N but was low to deficient in P, Mn, S, Fe, and B. The uptake of chemical elements into the foliage differed significantly according to whether the plants were growing as single or mixed species. When two species of *Grevillea* and grasses with evolutionary origins in low fertility soils were growing together, there was an enhanced uptake of P and Mn, in one or both species, in addition to other elements that were in low concentrations in the experimental soil. In contrast to this, *Protea neriifolia* that probably originated from a more fertile soil procured lesser amounts of the six elements from the soil when growing together with grasses. Two grasses tolerant of less fertile soils (*Dactylis glomerata* and *Poa cita*) obtained more nutrients when they grew together with proteas; this was a much stronger neighbour effect than was measured in *Lolium perenne* which is better adapted to high fertility soils. The findings illustrate both the functional compatibility and competition for plant nutrients in mixed-species rhizospheres. Species combinations substantially increased the acquisition of key elements from the soil nutrient pool.

## 1. Introduction

There is great variability in the mobility of chemical elements in the rhizospheres of different species that is reflected in the exploitation of the soil nutrient pool and the uptake of nutrients by plants [1,2]. This would suggest that combinations of plant species may be more effective than single species at modifying the mobility and management of chemical elements in soils [3]. The hypothesis underlying this study is that plants naturally adapted to low fertility or degraded soils are likely to benefit by growing in combination with other species that possess different functional traits in the rhizosphere, since most plant species have similar fundamental metabolic demands for the same range of key nutrients [4]. A strategy of sharing different capabilities to procure key soil nutrients might prevail over the competition for access to a limited resource. Contrary to this, plants adapted to more fertile soils may be more likely to employ a competitive strategy to rapidly acquire a majority share of nutrients from a more plentiful pool of available soil nutrients. We test this hypothesis by growing a combination of plants that do not naturally occur together, but that are known to have different root functional traits that are adapted to either fertile or infertile soils, combinations readily found both in the Poaceae (grasses) and Proteaceae (all species are referred to using the generic term ‘protea’ in the present paper). 

There are often added benefits to plant productivity from two or more plant species growing together. Intercropping in agriculture and horticulture provides increased yields, often referred to as transgressive overyielding [5,6,7,8,9], for example when legumes are grown together with other crops [10,11]. In this example, the fortuitous spillover of N fixed by rhizobial symbionts from the legume to neighbouring plants is generally viewed as incidental, even though it seems unlikely that evolution has favoured an adaptation in plants that expends metabolic energy and resources towards obtaining N that is then readily shared with competitors. Recently, we have shown that grasses reciprocate in this relationship by procuring key trace elements in the rhizosphere that are then passed on to legumes [3]; grass—clover assemblages enhanced overall productivity and uptake of P, K, S, Mn, Cu, Mo, and B. The present paper investigates whether the sharing of phosphorus and key trace elements can also be identified between proteas and grasses of different origins when they grow together in soil with sufficient N for healthy growth, but with deficiencies of other key nutrients. We selected plant species that are known to possess contrasting root functional traits, and which did not share a common biogeographical origin.

The uptake of chemical elements is known to differ between species and according to whether plants are grown in monoculture or in mixed species assemblages [12], or naturally found in the same location and habitat [3]. However, investigations of the physiological traits associated with uptake seldom extend to a consideration of the two-way sharing of soil nutrients. One of the few more detailed examples of complementarity is with plants that produce cluster roots, found within a few plant families, including the Proteaceae and in a few crops such as *Lupinus albus* (white lupin) [13,14]. Cluster roots primarily enable plants to exploit less labile pools of soil phosphorus (P) in P-deficient soils, by releasing organic acids to mobilise mineral P that is bound to metal cations and organic complexes in the soil [15]. There is some evidence that cluster roots can also facilitate the acquisition of nutrients by neighbouring plants [16]. Different and contrasting strategies to acquire soil nutrients are employed by grasses, for example through different root structures [17], using mycorrhizal associations [18], or by secreting organic acids (phytosiderophores) [19]. Our assumption was that the acquisition of P, which is most often the predominant element limiting plant growth, would play a definitive role in our findings. Complementarity has previously been found to explain coexistence between different functional groups of grasses [20]. 

The aim of this study was to investigate species complementarity in the context of P and key trace elements, by measuring the uptake of nutrients into the foliage of species of different origins growing together artificially in nutrient-depleted soil. The work is particularly relevant to the management of species diversity in low-fertility production systems, but also has potential significance for phytoremediation science and practice, where exotic species are introduced to contaminated and degraded soils to manipulate chemical elements [21,22,23].

## 2. Results

Elevated concentrations of P and Mn were particularly notable in the *Grevillea* spp. when grown with grasses (Figure 1), in contrast to *Protea neriifolia* foliage which had lower Mn concentrations when it was grown with the grasses. On at least one of the two sampling occasions, *Grevillea barklyana* had higher foliar concentrations of most chemical elements when growing together with grasses, particularly with *Dactylis glomerata* (cocksfoot) (Figure 2; Appendix A, Table A1). Only three elements in G. Robin Hood foliage differed significantly when it was growing with grasses (Figure 2; Appendix A, Table A1). Five of the elements with higher concentrations (P, Mg, S, Mn, Zn, and B) were nutrients known to be deficient in the soil, but also included K and Mo. In contrast, lower concentrations of 5 elements in addition to P, were often recorded in *P.*
*neriifolia* foliage when it had grown with grasses (Figure 3; Appendix A, Table A1). 

There were fewer changes in the elemental concentrations of the two pasture grasses, *Dactylis glomerata* (cocksfoot) and *Lolium perenne* (perennial ryegrass), when they were growing with proteas, and this was mainly related to higher concentrations of P, K, and Ca (Figure 4). *Poa cita* (a tussock grass) was different, with significantly elevated foliar P, K, Ca, S, Mn, and Zn when it was growing with proteas (Appendix A, Table A2). Focusing on *D. glomerata*, as the grass with the most evident neighbour effects (Figure 5), there was a shift in foliar nutrient concentrations of both proteas and grasses when they were growing with a neighbouring species, with a difference in nutrient uptake.

In terms of plant productivity, there were few significant differences in the biomass of the proteas or the grasses after they had grown with neighbouring plants, but the variability was high within each treatment (Figure 6). The exploitation of the soil pool of chemical elements may be more accurately represented by mass balance calculations (multiplication of dry wt. of foliage × nutrient concentration). This calculation was performed using data separately for each species (Figure 7; Appendix A, Table A3). More P, Mn, and Zn was procured by the *Grevillea* spp. but less by *P. neriifolia* when these species had grown in the same pots as the grasses. In percentage terms, neighbouring grasses only marginally reduced the biomass of protea foliage, but higher foliage concentrations of elements led to significantly increased total offtake of nutrients by as much as 100% in the proteas (Figure 8). There was a much lower biomass of grass foliage than of protea foliage, on average amounting to 11% of the latter, which meant that the total amount of each element extracted from the soil with the grasses was much less than the amount extracted by proteas (Appendix A, Table A3). Nonetheless, these calculations illustrated clear differences between *P. neriifolia* and the *Grevillea* spp., and between the different grass species (Figure 8). When the proteas and grasses were growing in combination with each other, compared to growing as monocultures, they extracted substantially increased amounts of nutrients from the soil.

## 3. Discussion

Measuring nutrient concentrations in foliage provides a surrogate but arguably the most realistic measure of the exploitation of the soil nutrient pool. In the present study, no direct attention was given to the processes in the soil that were responsible for differences in plant uptake. Interactions between plants that take place belowground are often overlooked, even though roots of different species are frequently intermingled, with growth, root exudates [24,25], root turnover [26], death, and decay [27] generally occurring in mixed species rhizospheres. Furthermore, the main input of nutrients to the soil is from root decomposition [28]. Otherwise, nutrients are largely bound to solid phase constituents of the soil with only small proportions entering the soil solution and becoming readily available for plant uptake. Undoubtedly, complex interactions in the rhizosphere affect soil biogeochemistry and nutrient acquisition by plants [24,29], but these were beyond the scope of the experimental work of the present study.

Our results showed that foliar concentrations of P and Mn uptake in proteas and grasses were consistently modified when they were grown together in species combinations, compared to when they were grown alone, but this also extended to elevated foliar concentrations of up to nine other elements (K, Ca, S, Fe, Mg, Zn, Cu, B, and Mo). Our assumption that P acquisition by the cluster-rooted proteas would play a definitive role is evident in the results, in terms of its increased P concentrations in the foliage of pasture grasses (*L. perenne* and *D. glomerata*), together with elevated K and Ca. The tussock grass (*P. cita*) had enhanced amounts of at least five elements (K, Ca, S, Mn, and Zn), but not of P, when growing with proteas. There appeared to be little reciprocation from the tussock grass to proteas. The low-fertility soil probably would have provided more natural conditions for this native grass, which is likely to explain its competitive abilities in procuring deficient nutrients from the soil when it was grown with proteas. 

*Dactylis glomerata* (cocksfoot) provided a better demonstration of species complementary than ryegrass, probably because it grew better and had more biomass. *Dactylis* provided the most significant impacts on nutrient uptake in the proteas. The two *Grevillea* species with origins in the ancient fertility-depleted soils of Australia benefited substantially through coexistence and the presence of neighbouring grasses in terms of an elevated uptake of most of the range of nutrients. In comparison, a lesser uptake of key nutrients (including Mn, Fe, Zn, and B) by the South African *Protea* when grown together with grasses implies some combination of competitive losses and less sharing, perhaps reflecting an evolutionary history on soils with more adequate levels of fertility, as discussed below. A recent study involved N and P fertilization of two fynbos sites in the Western Cape province of South Africa. Both study areas contained *P. neriifolia* as one of the two dominant species, although both sites had just been cleared of vegetation by wildfires [30]. Compared to six forest tree species, it was found that the thinner root traits of 12 emergent fynbos plants (that did not include proteas) provided a competitive advantage for the procurement of nitrogen, which unexpectedly appeared to be a more significant constraint than P. Nitrogen was largely overlooked in the present study as it appeared to be neither deficient nor a key element in our soil. Furthermore, proteas are not a natural component of vegetation in NZ soils.

There were differences between the *Protea* and the *Grevillea* spp. in terms of foliar nutrient concentrations when grown with grasses. The *Grevillea* spp. and grasses benefited by growing in combination with each other, with both obtaining more P, especially *G. barklyana*-*Dactylis* combinations. Otherwise, *Grevillea* also procured more S, Mg, Mn, Zn, and B, and the grasses obtained more K and Ca. *Protea neriifolia*-grass combinations were competitive rather than complementary, with the protea apparently less able to procure key deficient elements in the presence of grasses and had a higher foliar uptake of six elements when it was growing alone. There was no obvious benefit to *P. neriifolia* growing with neighbouring grasses. 

Likely explanations that would describe the processes responsible for different uptake patterns are to be widely found in the scientific literature. In broad terms, two of the most important ways that root exudates influence nutrient availability and uptake are through organic acid and phytosiderophore secretion [25]. In proteas, the availability of phosphorus in soil is the most important determinant of cluster root formation, and carboxylates exuded from the roots promote P mobilization in the soil [13]. Deficiencies of other elements, including N, K, Mn, and Fe also enhance cluster root development. Graminoid-secreted phytosiderophores release chelators to form complexes with soil metals, increasing metal solubility and mobility, particularly of Fe that is often in abundant but insoluble Fe (III) precipitates in soil. Many phenolics produced in the rhizosphere of dicots can form complexes with metals that may also increase their availability. In low-nutrient environments, plants can produce root exudates as symbiotic signals to soil microbes involved in nutrient procurement, to use extracellular enzymes to release P from organic compounds, and organic acids to solubilize soil Ca, Fe, and Al phosphates [31]. There is increasing evidence that plants can be complementary to one another to procure nutrients more efficiently [32] and at reduced metabolic costs [33]. However, the mechanistic explanations are complex; for example, many phytohormones are involved in interactions between roots, soil, and microbial communities [34]. Rhizosphere processes are insufficiently understood [35], and there remains a paucity of studies that provide mechanistic evidence from soil-based systems [36].

The importance of considering multiple nutrient constraints on plant productivity has been stressed elsewhere [37]. The requirements of plants for similar base concentrations but differing amounts of particular nutrients are likely to be specific to the plant species, and this is probably reflected in the differing foliar concentrations recorded in the present study. The most likely capacity-based approach to nutrient acquisition [29] assumes that plants expend metabolic energy to acquire nutrients by exploiting gradients of nutrient molecules inside and outside the root, using specific nutrient-acquiring proteins, pumps, transporters, and channels [29].

There was some evidence from the present study of transgressive overyielding in the context of an increased proportion of the key nutrients being removed from the total soil pool by combinations of species compared to monocultures. No account was taken of nutrient uptake into the woody or green stems of the proteas in terms of total offtake. Nutrient concentrations in these plant components would be expected to be much lower than in foliage, but the amount of additional nutrients in these fractions could have great relevance to production systems and phytoremediation technologies. The competition for nutrients, facilitation, and complementarity are all major driving forces of ecosystem productivity [35]. In the context of species complementarity, we have shown that species not naturally found together have functional attributes in the rhizosphere that can be shared to facilitate an improved procurement of nutrients. 

A better understanding of these functional traits could be very useful in the context of the sustainability of plant communities of native or exotic species, or combinations of both (novel native plant communities) in New Zealand and elsewhere [3,22]. This could be a step towards the better management of vegetation and soils in low-input agricultural systems. More fundamental and applied research knowledge of functional biodiversity and plant species complementarity is required in the context of soil biogeochemistry. The findings of the present study illustrate functional compatibility as well as competition between plant rhizospheres for plant nutrients. Beneficial coexistence appears to be explained by the differences between the plant rhizospheres of different species which exploit different components of the soil nutrient pool [38,39]. This implies that the enhancement of species diversity, for example, beyond simply focusing on legumes and grasses in pasture agronomy [3], may be a better way to manage ecosystems, including production systems, with low-fertility or degraded soils. We suggest that it would be worthwhile to extend the experimental approach used in the present study to a wider range of species combinations that have a direct practical application to less-intensive grazing systems, phytotechnologies, and to the conservation and restoration of biodiversity.

## 4. Materials and Methods

Soil (1–20 cm depth) was collected from a site in Canterbury (altitude 611 m), South Island, New Zealand (S 43°20’35”, E 171°36’59”), that was described in detail in earlier papers [3,40]. The site was originally forested, probably until the mid-19th century. Since then, the land has been extensively grazed by sheep and wild ruminants but otherwise has been largely undisturbed. Undoubtedly, the soil, which had patchy vegetation cover, was substantially degraded through forest removal, mammalian grazing impacts, exposure, and erosion, probably for a century or more. The collected soil was thoroughly mixed, then air-dried and sieved (2 mm) prior to being used in the experimental work. Samples were analysed using standard methods by Analytical Services, Soils and Physical Sciences Department, Lincoln University (Table 1), showing a range of key determinants (pH, Ca, sulphate-S, soluble P, Cu, Mg, Mn, and B) were less than optimum for plant growth. Available P, Ca, and B were extremely deficient, although there was adequate N for healthy plant growth at yields that could be achieved in the landscape of its origin.

Species were selected from the Proteaceae (proteas) and Poaceae (grasses) as being representative of functional groupings that are known to possess different and contrasting traits of nutrient acquisition in the rhizosphere. We grew three species of proteas: *Grevillea barklyana* F. Muell. Ex Benth. (Gully- or large-leaf grevillea) endemic to south-western Australia; *Grevillea* Robin Hood (a hybrid cultivar of *G. hookeriana* Meisn.), endemic to south-eastern Australia; and *Protea neriifolia* R. Br. (narrow-leaf sugarbush), endemic to the Western and Eastern Cape of South Africa. Fynbos soils are characterised by low soil fertility [41], but dense stands of *P. nerriifolia* are naturally found on less-leached granite-derived renosterveld soils on mountain slopes [42,43] that are more fertile [44]. The inherent fertility is likely to be much lower in the more ancient and strongly weathered soils of Australia [30,45] than in the South African soils. All three species of protea produce cluster roots. The three species only grow ornamentally in New Zealand. These proteas were grown either alone or in combination with single species of grasses: one of two widespread and common grasses, *Lolium perenne* L. (perennial ryegrass), and *Dactylis glomerata* L. (cocksfoot), or *Poa cita* Edgar (silver tussock) which is an endemic New Zealand tussock grass [46]. 

A pot trial was set up in a glasshouse at Lincoln University. The cluster root forming species were taken from cuttings of single plants, rooted in seed trays, and then transplanted into 3.5 L plastic pots (15 cm diameter, 20 cm height). Thirteen experimental treatments consisted of three cluster-root forming species and three grasses growing either singly or in combinations, with five replicates per treatment. Pots were arranged in a randomized single block design on a glasshouse bench. Glasshouse temperatures for the duration of the experiment were 19.0 °C (mean); 13.6 °C (min)–34.7 °C (max). Plants were watered sparingly every two days. Survivorship was generally good, but one Gb and one Pn died after transplanting, one Pn (Rg) died after first harvesting, and one Pn (Tg) died a few days before the final harvest. 

After 6 months’ growth, plant material was sampled for chemical analysis; five leaves were harvested from each of the proteas, and the grasses were harvested to 2 cm above the soil surface. This was repeated 6 months later when the plants were completely harvested, and root systems were separated by careful washing. Aboveground plant material was sorted into separate species, dried (65 °C, 48 h), weighed, and finely ground, microwave digested in 5 M HNO_3_, and then chemically analysed by ICP-OES using standard methodology. For statistical analysis, data not normally distributed were log-transformed before analysis. Differences between means were determined using one-way ANOVA, with a post-hoc Fisher LSD test. All analyses were conducted using Minitab 19.

## 5. Conclusions

This study provided evidence of the compatibility between plant species with contrasting functional rhizosphere traits. Nutrient constraints in the experimental soil were better addressed by combinations of proteas and grasses growing together. Species combinations substantially increased the procurement of key deficient elements, providing evidence that mixed-species rhizospheres enable an improved exploitation of the soil nutrient resource. This was the result of higher foliar nutrient concentrations and enhanced total uptake of nutrients. The sharing of access to the soil nutrient pool is evident in these findings. This implies that a strategy of competition for plant nutrients may be less important than functional compatibility and mutual enhancement of uptake between neighbouring species in low-fertility or degraded soils. 

## Figures and Tables

**Figure 1 plants-11-01370-f001:**
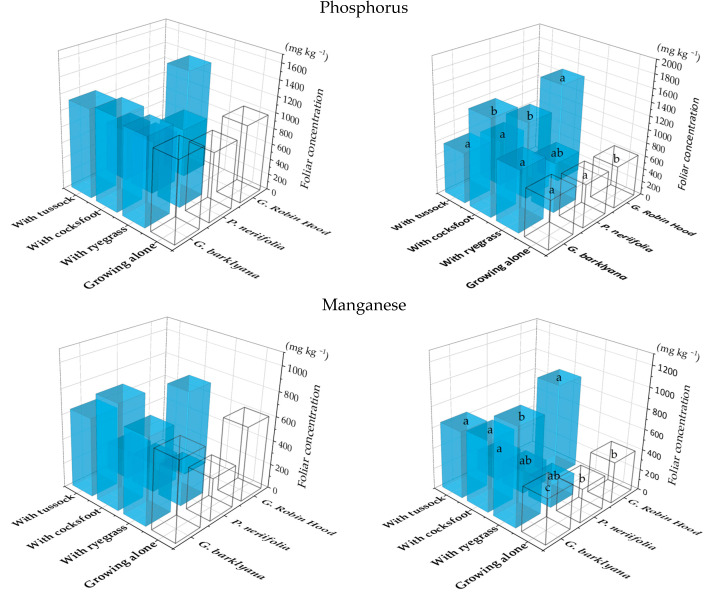
Phosphorus and manganese concentrations in the foliage of the three species of proteas when they were growing alone (open bars) or with one of three species of grass: *Lolium perenne* (ryegrass), *Dactylis glomerata* (cocksfoot), or *Poa cita* (tussock grass) (coloured bars). Charts show results of 1st (LHS) and 2nd (RHS) sampling. Different letters each indicate significant differences (*p* < 0.05) for each protea (full results in Appendix A, Table A1).

**Figure 2 plants-11-01370-f002:**
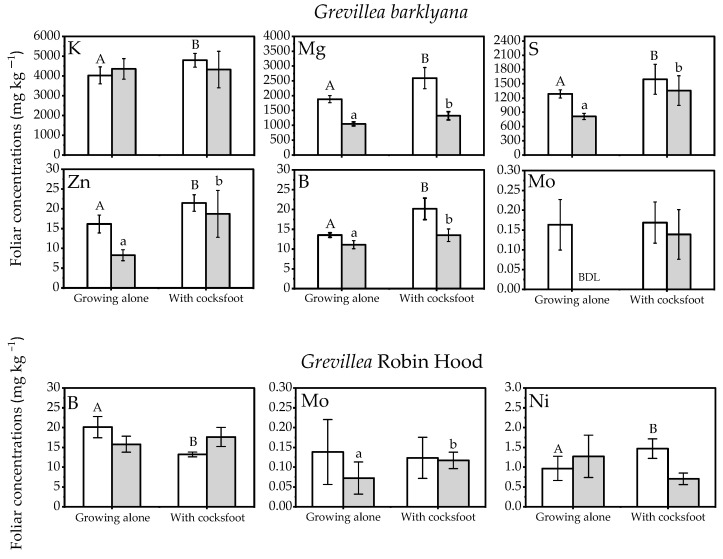
Nutrient concentrations in the foliage of the two *Grevillea* spp. when they were growing alone or together with *Dactylis glomerata* (cocksfoot) (open bars, first sampling; shaded bars, final sampling). Different letters each indicate significant differences (*p* < 0.05) within each sampling event (upper case letter indicate differences at the first sampling, lower case letters indicate significant differences at final sampling). Elements without significant differences are not shown (full results in Appendix A, Table A1).

**Figure 3 plants-11-01370-f003:**
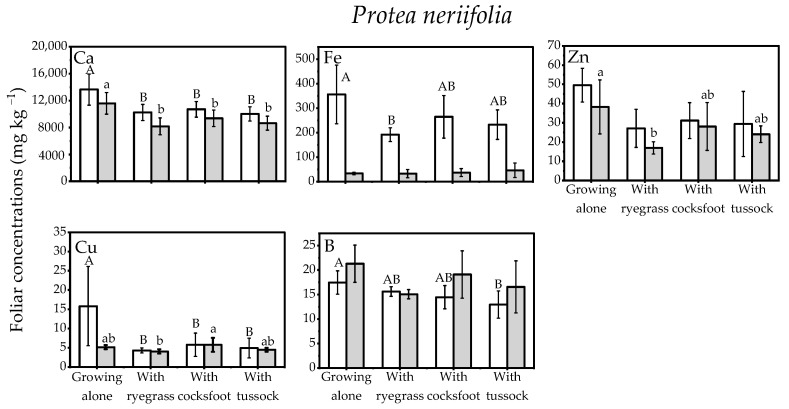
Nutrient concentrations in the foliage of the *Protea neriifolia* when grown alone or together with the three species of grass: *Lolium perenne* (ryegrass), *Dactylis glomerata* (cocksfoot), or *Poa cita* (tussock grass) (Open bars, first sampling; Shaded bars, final sampling). Different letters each indicate significant differences (*p* < 0.05) within each sampling event (upper case letters indicate differences at the first sampling, lower case letters indicate significant differences at final sampling). Elements without significant differences are not shown (full results in Appendix A, Table A1).

**Figure 4 plants-11-01370-f004:**
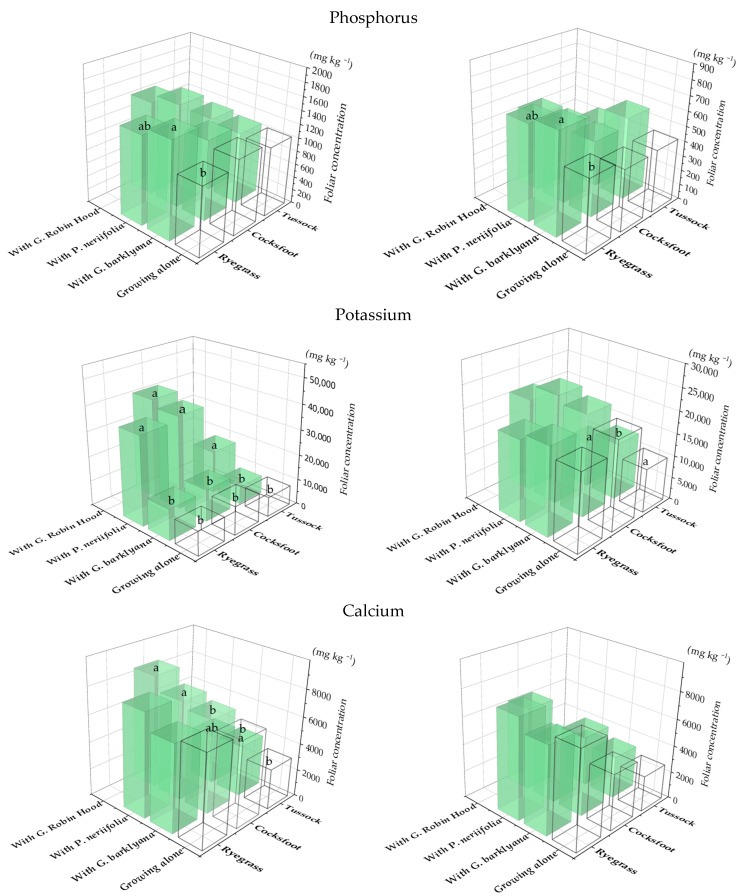
Phosphorus, Ca, and Mn concentrations in the foliage of the three species of grass (*Lolium perenne*, ryegrass; *Dactylis glomerata*, cocksfoot; *Poa cita*, tussock grass) when they were growing alone (open bars) or with one of three species of proteas (coloured bars). Charts show 1st (LHS) and 2nd (RHS) sampling. Different letters each indicate significant differences (*p* < 0.05) each for each species of grass (full data in Appendix A, Table A2).

**Figure 5 plants-11-01370-f005:**
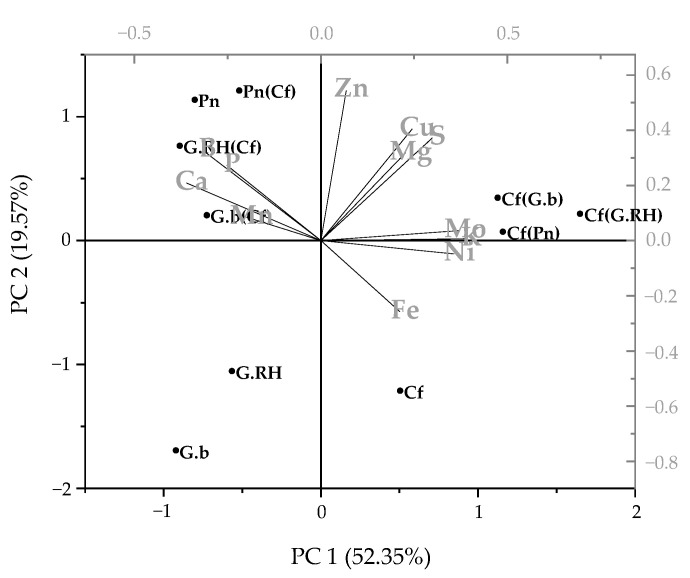
Principal Components Analysis describing foliar nutrient concentration data for each of the protea species (Gb, *Grevillea barklyana*; *Protea neriifolia*, Pn; Grevillea Robin Hood, G.RH) growing alone or with *Dactylis glomerata* (cocksfoot, Cf), and for cocksfoot growing alone or with each of the proteas. Abbreviations in brackets indicate the companion species.

**Figure 6 plants-11-01370-f006:**
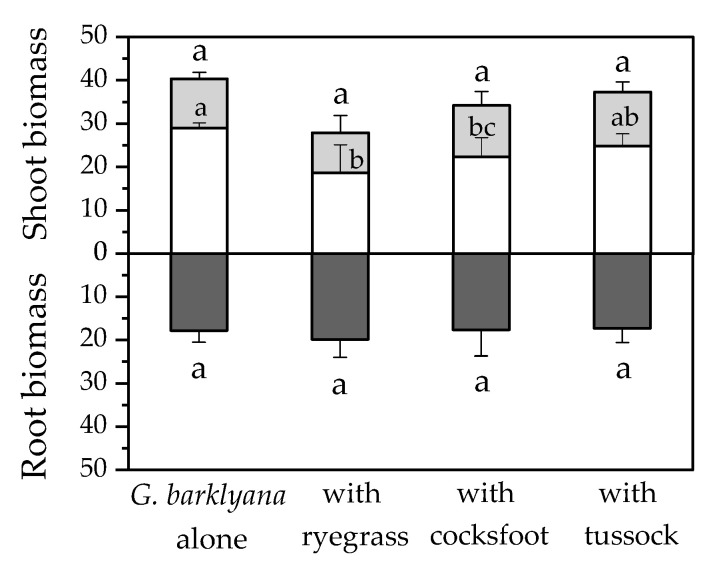
Harvested biomass of *Grevillea barklyana* (dry wt., g pot^−1^) when it was grown alone or with one of the three species of grass: *Lolium perenne* (ryegrass), *Dactylis glomerata* (cocksfoot), or *Poa cita* (tussock grass). Shoot biomass shows stems (open bars) and foliage (shaded bars). Different letters indicate significant differences (*p* < 0.05) within separate plant components.

**Figure 7 plants-11-01370-f007:**
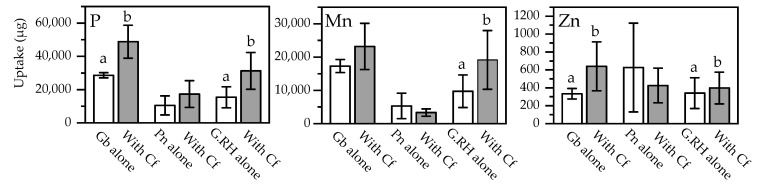
Total uptake of P, Mn, and Zn into foliage of each of the three species of Protea, when they were growing alone or with *Dactylis glomerata* (cocksfoot): Gb: *Grevillea barklyana*, Pn: *Protea neriifolia*, G.RH: Grevillea Robin Hood, Cf: Cocksfoot. Open bars, first sampling; shaded bars, final sampling. Different letters each indicate significant differences (*p* < 0.05) within each of the sampling events. Elements without significant differences are not shown (full data in Appendix A, Table A3).

**Figure 8 plants-11-01370-f008:**
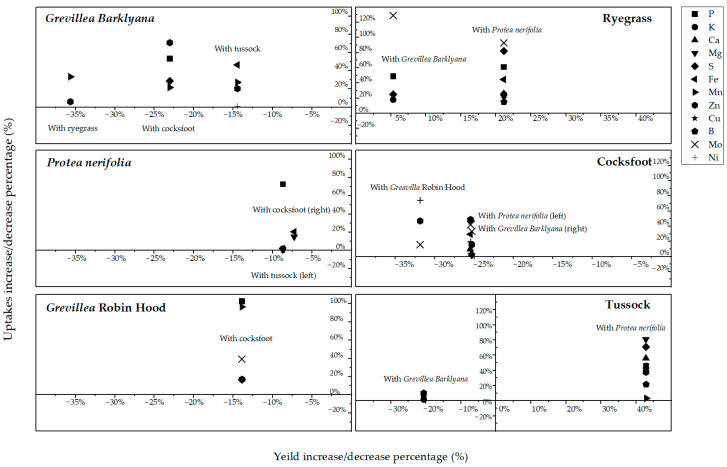
The percentage change in yield (horizontal axes) and total foliar uptake of key nutrients (vertical axes), when each of the protea and grass species (*Lolium perenne*, ryegrass; *Dactylis glomerata*, cocksfoot; *Poa cita*, tussock grass) was grown with a companion species. Percentage change is the difference to when each species grew alone.

**Table 1 plants-11-01370-t001:** Analysis of physico-chemical determinants in the experimental soil.

Indicators	Units	Concentration	Typical Range *
**pH ^1^**	pH Units	5.70	5.70–6.20
**Total Nitrogen ^2^**	%	0.46	0.30–0.60
**Total Carbon ^3^**	%	5.80	-
**Organic Matter ^4^**	%	10.0	7.00–17.0
**Total Phosphorus**	mg kg^−1^	464	700–1600
**Olsen Phosphorus ^5^**	mg L^−1^	4.33	20.0–30.0
**Potassium ^6^**	me/100 g	0.49	0.30–0.60
**Calcium ^6^**	me/100 g	2.03	5.00–12.0
**Magnesium ^7^**	me/100 g	0.60	0.60–1.20
**Sodium ^7^**	me/100 g	0.05	0.00–0.30
**Sulphate Sulphur ^8^**	mg kg^−1^	6.43	10.0–20.0
**Iron ^7^**	mg L^−1^	84.0	500–1000
**Manganese ^7^**	mg L^−1^	3.20	8.00–65.0
**Zinc ^7^**	mg L^−1^	1.75	0.80–4.00
**Copper** **^7^**	mg L^−1^	0.37	0.40–2.00
**Boron ^7^**	mg L^−1^	0.19	0.60–1.20

* Typical range for agricultural soils in New Zealand. Analyses follow standard methodology from a commercial laboratory. Analyses by the commercial laboratory were routinely carried out by defined volume rather than mass of soil. Method: ^1^ 1:2 (*v*/*v*) soil:water slurry followed by potentiometric determination of pH. ^2^ Determined by NIR, calibration based on total N by Dumas combustion. ^3^ Determined by NIR, calibration based on total Carbon by Dumas combustion. ^4^ Organic Matter is 1.72 × Total Carbon. ^5^ Olsen extraction followed by Molybdenum Blue colorimetry. ^6^ 1 M Neutral ammonium acetate extraction followed by ICP-OES. ^7^ Mehlich 3 Extraction followed by ICP-OES. ^8^ 0.02 M Potassium phosphate extraction followed by Ion Chromatography.

## Data Availability

Not applicable.

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
