# Peer review of "Plant Species Complementarity in Low-Fertility Degraded Soil"

_plants, 2022, doi:10.3390/plants11101370_

Round 1
Reviewer 1 Report
The manuscript entitled 'Plant Species Complementarity in Low-Fertility Degraded Soil' describes a study on species complementarity and soil nutrient acquisition between paired combinations of Proteacea species and grass species.
An important point which need to be addressed.
- It was stated in line 199-202 that Protea neriifolia had less uptake of key nutrients (including Mn, Fe, Zn and B) when growing together with grasses implies some combination of competitive losses and less sharing, perhaps reflecting an evolutionary history on soils with more adequate fertility.
It would be worthwhile to explain a bit more why this could so because my reading of the literature indicates that the Fynbos biome where P. neriifolia occurs is generally considered one of the most phosphorus-limited plant communities worldwide. It perhaps can tolerate this competition with grasses for nutrients.
It would be better to use Proteaceae instead of Proteas as Grevilleas are not Proteas.
Author Response
The reviewer has provided two interesting comments to which we have give considerably more thought and made appropriate amendments.
Regarding the low fertility of fynbos soils, we have modified the section in Materials and Methods, which now reads (Lines 293-297): "fynbos soils are characterised by low soil fertility [41], but dense stands of P. nerriifolia are naturally found on less-leached granite-derived renosterveld soils on mountain slopes [42,43] that are less infertile [44]. The inherent fertility of these South African soils is likely to be much lower in the more ancient and strongly weathered soils of Australia [30, 45]." [NB. Four of the references cited here have been added to the reference list]
With regard to the use of the generic term 'protea' to include the Grevilleas, we think this makes the style of the paper much easier to read, using a common name as we have for grasses. We would like to retain the use of this term, but we have added the following section to the end of the first paragraph of the Introduction (Lines 41-41): "Proteaceae (all species are referred to using the generic term ‘protea’ in the present paper)."
Reviewer 2 Report
Review report for the article
Plants-1681174
The manuscript entitles “Plant Species Complementarity in Low-Fertility Degraded Soil” has been written good and have some comments below:
Line no 49, Sentence recorrect and start with Recently, it is observed that in spite of We have recently shown.
Line no 89. What do you mean about G. Robin Hood. I could not understand why authors quote Figure 3 before the figure 2 inside the text??
Line no 86, What do you mean about this sentence “ Five of the elements with higher concentrations (P, Mg, S, Mn, Zn, B) were 86 nutrients known to be deficient in the soil, in addition to elevated K and Mo” please rewrite it.
In Figure 2, please explain what do you mean the word, A, B, a, and b?
In methodology, How many plants grown for the study?
I couldnot understand as authors did very well work inside the manuscript but not used any statistical programme, why?
Authors need to explain and interpretate the results with efficient statistical programme and justify the results are significant or not.
How did the authors find results are significantly proved?
In Tables, Author have written the Different letters separately indicate significant differences (p < 0.05), Could you please explain which test you have used for this study?
I am wonder to see the manuscript that authors not given any conclusion in the end. Please write conclusion of this study.
References should be according to the journals guideline. And follow same format for all the references.
English language should be improved in throughout the manuscript.

Author Response
We think we have addressed ALL of this reviewers specific comments and we have made improvement to the sections the reviewer considered needed attention as in the tick-boxes. We are especially grateful that the missing Table 1 was picked up - inexcusably this somehow disappeared from the submitted ms. Table 1 has now been included. Each comment has been addressed as follows:
Line no 49, Sentence recorrect and start with Recently, it is observed that in spite of We have recently shown. [This is corrected]
Line no 89. What do you mean about G. Robin Hood. I could not understand why authors quote Figure 3 before the figure 2 inside the text?? [This is corrected]
Line no 86, What do you mean about this sentence “ Five of the elements with higher concentrations (P, Mg, S, Mn, Zn, B) were 86 nutrients known to be deficient in the soil, in addition to elevated K and Mo” please rewrite it. [This is corrected and explained properly]
In Figure 2, please explain what do you mean the word, A, B, a, and b? [This is now explained properly in each legend]
In methodology, How many plants grown for the study? [This is explained in detail in Lines 304-306]
I couldnot understand as authors did very well work inside the manuscript but not used any statistical programme, why? [Thank you for noticing this omission: it is now added in Lines 317-320]
Authors need to explain and interpretate the results with efficient statistical programme and justify the results are significant or not. [As above, with apologies for previous diffculty to understand this]
How did the authors find results are significantly proved? [Ditto]
In Tables, Author have written the Different letters separately indicate significant differences (p < 0.05), Could you please explain which test you have used for this study? [Ditto]
I am wonder to see the manuscript that authors not given any conclusion in the end. Please write conclusion of this study. [We have checked recent papers in the journal and it seems most common to include the conclusions as the last paragraph of the Discussion, as we have done. We are however happy for the publisher to insert a Conclusion sub-title before the last paragraph if that is the current preference]
References should be according to the journals guideline. And follow same format for all the references. [They are fully corrected]
English language should be improved in throughout the manuscript. [Three of the authors are very well published native English speakers. The corresponding author led the writing of the final draft of the manuscript. We have corrected a few small slippages, for which we apologize, but otherwise we are confident the standard of written English is accordant with our usual standards]
Round 2
Reviewer 2 Report
Dear Authors
I am happy that you have improved manuscripts as per the suggestions. Therefore, in my opinion, now paper may be published in this journal.
Author Response
This does not require a response, but I have responded to the comments of the Academic Editor